

# Forecasting the COVID-19 transmission in Italy based on the minimum spanning tree of dynamic region network

Min Dong[1], Xuhang Zhang[1], Kun Yang[1], Rui Liu[2,3] and Pei Chen[2]

[1] School of Computer Science and Engineering, South China University of Technology, Guangzhou, China
[2] School of Mathematics, South China University of Technology, Guangzhou, China
[3] Pazhou Lab, Guangzhou, Guangdong, China

## ABSTRACT

**Background.** Italy surpassed 1.5 million confirmed Coronavirus Disease 2019 (COVID-19) infections on November 26, as its death toll rose rapidly in the second wave of COVID-19 outbreak which is a heavy burden on hospitals. Therefore, it is necessary to forecast and early warn the potential outbreak of COVID-19 in the future, which facilitates the timely implementation of appropriate control measures. However, real-time prediction of COVID-19 transmission and outbreaks is usually challenging because of its complexity intertwining both biological systems and social systems.

**Methods.** By mining the dynamical information from region networks and the short-term time series data, we developed a data-driven model, the minimum-spanning-tree-based dynamical network marker (MST-DNM), to quantitatively analyze and monitor the dynamical process of COVID-19 spreading. Specifically, we collected the historical information of daily cases caused by COVID-19 infection in Italy from February 24, 2020 to November 28, 2020. When applied to the region network of Italy, the MST-DNM model has the ability to monitor the whole process of COVID-19 transmission and successfully identify the early-warning signals. The interpretability and practical significance of our model are explained in detail in this study.

**Results.** The study on the dynamical changes of Italian region networks reveals the dynamic of COVID-19 transmission at the network level. It is noteworthy that the driving force of MST-DNM only relies on small samples rather than years of time series data. Therefore, it is of great potential in public surveillance for emerging infectious diseases.

## INTRODUCTION

The world is currently witnessing a major and devastating pandemic with substantial mortality and morbidity–Coronavirus Disease 2019 (COVID-19) (*Mohanty et al., 2020*). It was declared by the World Health Organization (WHO) as a public health emergency of international concern in January 2020 (*Team, 2020*; *Lai et al., 2020*). As of November 28, 2020, about 60 million cases and 1.45 million deaths were confirmed globally. Italy is one of the most affected countries. In about two months, i.e., from mid-February 2020 to

Corresponding authors
Rui Liu, scliurui@scut.edu.cn
Pei Chen, chenpei@scut.edu.cn

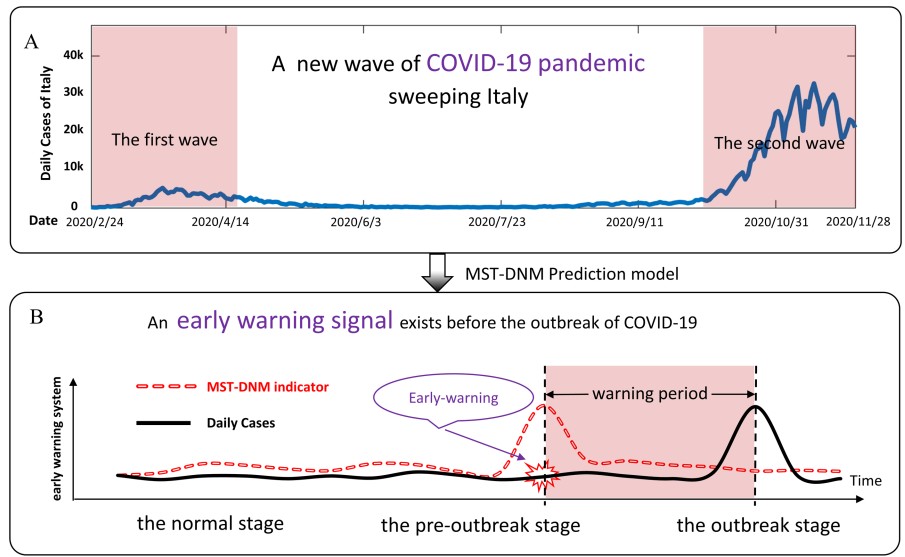

**Figure 1** **Schematic diagram of COVID-19 pandemic in Italy and prediction principle.** (A) From mid-February 2020 to mid-April 2020, the epidemic in Italy reached its first peak. Since then, the epidemic curve gradually declined until early October, and then the spread of COVID-19 infection accelerated again until today. A new wave of COVID-19 outbreaks is affecting Italy. (B) According to the MST-DNM model, the process of COVID-19 outbreaks is divided into three stages, including the normal stage, the pre-outbreak stage and the outbreak stage. The sudden increase in the MST-DNM indicator indicates a transition from the normal stage to the pre-outbreak stage, i.e., the critical point before the upcoming outbreak of COVID-19 that results in an increase in daily cases.

mid-April 2020, it has been one of the main epicenters of the COVID-19 pandemic when the epidemic reached its first peak. Then, the epidemic curve gradually decreased until early October 2020, after which the spread of infection accelerated again until today (*Perone, 2020*). As shown in Fig. 1A, a new and more severe epidemic is sweeping Italy. As of November 28, Italy had suffered 54,363 deaths and 1,564,532 cases (*Dong, Du & Gardner, 2020*). Therefore, there is an urgent need for an effective and low-cost model to build an epidemic surveillance system to help countries severely affected by the epidemic like Italy to warn of a new wave of COVID-19 outbreaks in the future.

The COVID-19 pandemic has sparked an intense debate about the factors underlying the dynamics of the outbreak (*Pacheco Coelho et al., 2020*; *Pequeno et al., 2020*). Meanwhile, the study of mathematical models of epidemiology is helpful to understand the dynamics of epidemics, being an important tool to evaluate the potential effects of preventive and controlled measures, especially when their characteristics are still unclear (*Chatterjee et al., 2020*; *Khan et al., 2019*; *Cheng et al., 2020*). Under such circumstances, by exploring the dynamical information from region networks and time-series data, we employed a combined model, the minimum-spanning-tree-based dynamical network marker (MST-DNM) (*Yang et al., 2020*), to quantitatively describe the dynamics of COVID-19 transmission and thus identify the early-warning signal of a new wave of COVID-19 pandemic in Italy. This is the first time that this model has been applied to the study of the emerging infectious disease like COVID-19. This model is improved from our

recently proposed concept, the so-called dynamical network biomarker (DNB), which determines the critical state of complex diseases by analyzing the dynamics of driven biomolecules (i.e., a group of genes and proteins that are the leading factors to the critical state transition) (*Chen et al., 2012*). The DNB based methods have been applied to a number of biological progresses and obtained remarkable results, including identifying the critical points of cell fate decision (*Mojtahedi et al., 2016*) and cellular differentiation (*Richard et al., 2016*), detecting the critical period during various biological processes (*Liu et al., 2014*; *Liu et al., 2019*; *Chen et al., 2015*; *Chen et al., 2017*), and predicting the warning signals of influenza outbreaks (*Yang et al., 2020*; *Chen et al., 2019*; *Chen et al., 2020*).

By extracting the minimum spanning tree from the dynamic region network, we could typically describe the dynamics of the spread of the infectious diseases among regions. Specifically, we collected the historical information of daily cases by COVID-19 infection in Italy from February 24, 2020 to November 28, 2020. When applied to the region networks constructed based on geographical location and traffic conditions, the MST-DNM model has the ability to monitor the whole process of epidemic spread in Italy, and successfully identify the early-warning signals about two weeks in advance. It is worth noting that in the previous research, the key role of the minimum spanning tree in this model has been described in detail (*Yang et al., 2020*), which can avoid issuing wrong warning signals due to the appearance of local abnormal correlation. Therefore, we pay more attention to the interpretability and practical significance of the MST-DNM model in our study, as detailed in the Materials & Methods section and in the Results section. Consequently, our model is quite suitable for predicting the potential outbreak of COVID-19 in Italy with the characteristics of nonlinearity time series and small sample size based on Italian region network, which may help to develop new control strategies for COVID-19 before its new wave of outbreaks.

## MATERIALS & METHODS

### Theoretical background

The spread of infectious diseases in a region is described as the dynamic evolution of a nonlinear system, while the outbreak of COVID-19 is regarded as a qualitative state transition of a dynamic system (*Yang et al., 2020*). From the perspective of dynamic modelling, the stage before the COVID-19 outbreak is regarded as a pre-outbreak stage, immediately after which the system undergoes a critical transition. Then the dynamical process of an epidemic system can be roughly modelled as three stages similar to the dynamics of disease progression (*Chen et al., 2012*): (i) the normal stage, which is a stable stage with high resilience; (ii) the pre-outbreak stage, which is dynamically unstable. At this stage, the epidemic is still controllable through appropriate measures; and (iii) the outbreak stage, which is an uncontrolled stage with high elastic dynamic features. As shown in Fig. 1B, when the system transits from the normal stage to the pre-outbreak stage, the region network changes significantly and the indicator of our model rises sharply. Different from the traditional detection of the outbreak stage, our model could determine the pre-outbreak stage which generally has no obvious abnormalities but with high potential of state transition into a severe and irreversible stage.

## Data collection

Data about the COVID-19 pandemic can be obtained from the GitHub repository managed by Johns Hopkins University for modeling (*Dong, Du & Gardner, 2020*), which contains publicly available data from multiple sources. The open dataset of Italy we use is available on the website COVID-19 in Italy at: https://github.com/pcm-dpc/COVID-19. This database has been created and managed by the Italian Civil Protection Department, which is updated and integrated daily (*Italian Civil Protection D et al., 2020*).

## Algorithm applied in Italy

The MST-DNM model is illustrated in Fig. 2 and the detailed process of our model applied in Italy is described in the following four steps.

### i. *Modeling and mapping*

It is noted that our model is applied to the region network to monitor the COVID-19 transmission and outbreak in Italy. Therefore, it is necessary to construct the regional network based on the Italian regions' geographic distribution and their adjacent information at first. The adjacent information is shown in Table S1 of Supplementary Information. In the network, each node represents a region or an autonomous province, while each edge represents the adjacent relation between two regions. Then a $21 * 279$ data matrix formed by the records of COVID-19 daily confirmed cases is mapped to the network. The region network model of Italy is presented as in Fig. 3.

### ii. *Weighting and extracting*

The region network in the first step can be represented as an undirected graph $G = (V, E)$, which contains a collection $V = \{v_i\}_{i=1}^{M}$ composed of $M$ vertexes and a collection $E = \{e_j\}_{j=1}^{N}$ composed of $N$ edges in this network. In the Italian region network (Fig. 3), $M$ is 21 and $N$ is 34. It should be noted that in order to make the application of our model easier to understand, Node 15 is not considered in the subsequent calculations. When the model is applied to other regions, the construction of regional network should be reconsidered, instead of simply excluding islands. In addition, as shown in Fig. S1, as the COVID-19 epicenter is located in northern Italy, whether Sardinia is included in the model has little influence on the final warning results.

The number of daily confirmed cases of a region is considered as a sample $s$ to form a set of time series data. Therefore, for each vertex $v_i$ of the region network on the day $t$, there is a corresponding time series of confirmed cases $S_t^{v_i} = \{s_1, s_2, \ldots, s_t\}$. In order to assign a weight $W_t^k$ to each edge $e^k$ of the region network on the day $t$, it is necessary to calculate the correlations between the two vertexes $v_i, v_j$ of the edge $e^k$ as follows:
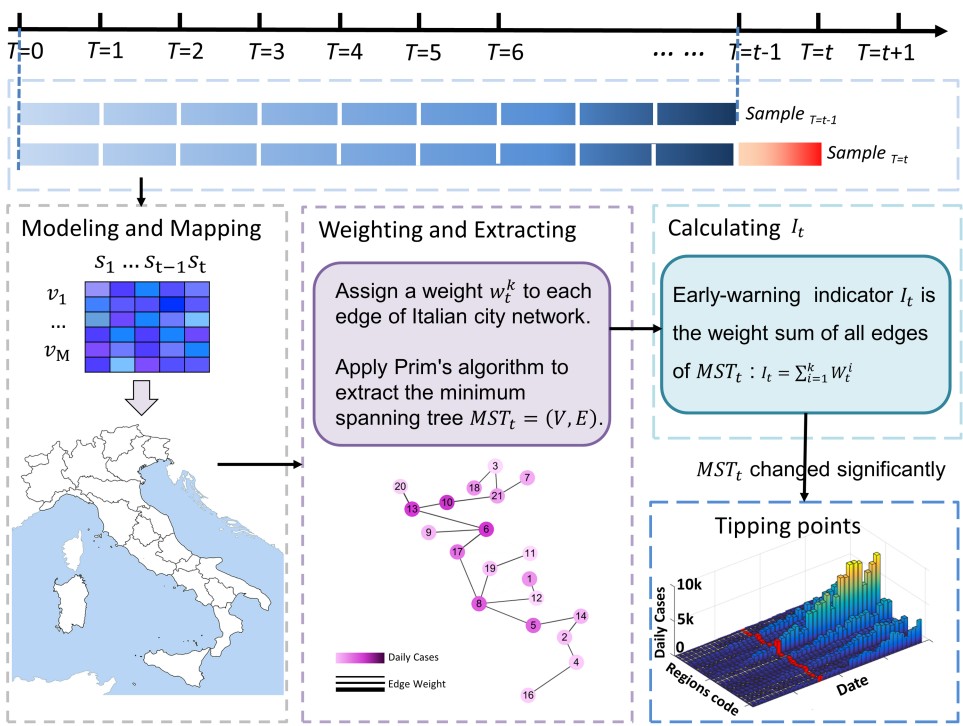

**Figure 2   The flow chart of algorithm application in Italy.** The flow chart above shows how the algorithm works in Italy based on region networks and minimum spanning tree. The model is driven by time series and dynamic region networks over time. Regarding a point $T = t$ as a candidate tipping point, MST-DNM indications can be calculated based on weighted region networks. If the indication increases significantly, the candidate $T = t$ is determined as the identified tipping point, and the algorithm ends. Otherwise, if there is no significant change, then $T = t$ is classified as a time point belonging to the normal stage, and the algorithm continues with $T = t+1$ being a new candidate tipping point.

$$W_t^k = \left|\Delta SD_t\left(i,j\right)\right| * \left|\Delta PCC_t\left(i,j\right)\right|, \tag{1}$$

where

$$\Delta SD_t\left(i,j\right) = \left|\overline{SD_t\left(S_t^{v_i}\right) + SD_t\left(S_t^{v_j}\right)} - \overline{SD_{t-1}\left(S_{t-1}^{v_i}\right) + SD_{t-1}\left(S_{t-1}^{v_j}\right)}\right| \tag{2}$$

is the differential standard deviation of the nodes $v_i, v_j$ on day $t$ and $t-1$, and $SD_t\left(S_t^{v_i}\right)$ and $SD_t\left(S_t^{v_j}\right)$ represent the standard deviation of the time series data of the two vertices $v_i, v_j$.

$$\Delta PCC_t\left(i,j\right) = \left|PCC_t\left(S_t^{v_i}, S_t^{v_j}\right)\right| - \left|PCC_{t-1}(S_{t-1}^{v_i}, S_{t-1}^{v_j})\right| \tag{3}$$

is the differential Pearson's correlation coefficient between the two vertices $v_i, v_j$ of the edge $e^k$, where $PCC_t\left(S_t^{v_i}, S_t^{v_j}\right)$ and $PCC_{t-1}(S_{t-1}^{v_i}, S_{t-1}^{v_j})$ represent the Pearson's correlation coefficient between the two vertices $v_i, v_j$ on day $t$ and $t-1$ respectively.
A

B

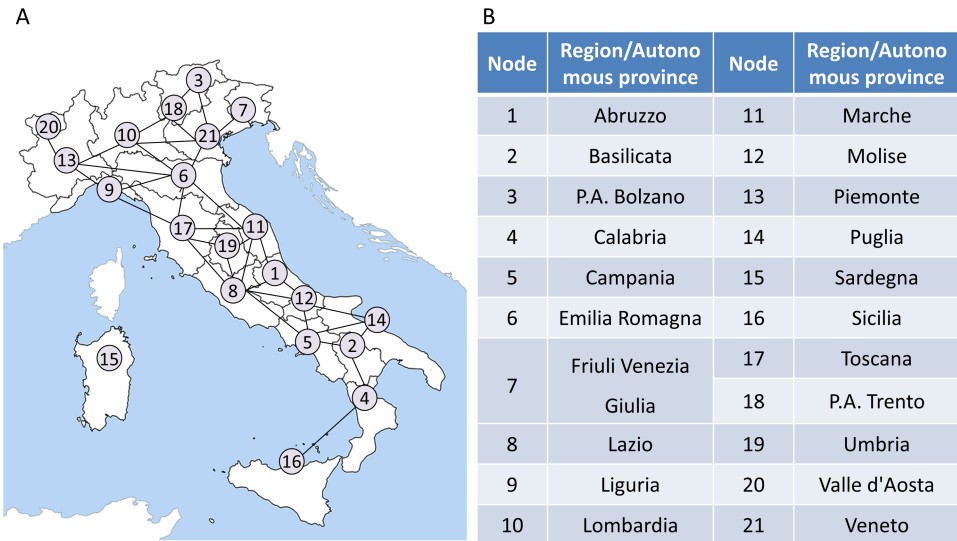

| Node | Region/Autonomous province | Node | Region/Autonomous province |
|------|-----------------------------|------|-----------------------------|
| 1 | Abruzzo | 11 | Marche |
| 2 | Basilicata | 12 | Molise |
| 3 | P.A. Bolzano | 13 | Piemonte |
| 4 | Calabria | 14 | Puglia |
| 5 | Campania | 15 | Sardegna |
| 6 | Emilia Romagna | 16 | Sicilia |
| 7 | Friuli Venezia Giulia | 17 | Toscana |
| | | 18 | P.A. Trento |
| 8 | Lazio | 19 | Umbria |
| 9 | Liguria | 20 | Valle d'Aosta |
| 10 | Lombardia | 21 | Veneto |

**Figure 3** **The region network model of Italy.** (A) A 21-node network model is constructed based on the geographic information and adjacent relationships of 21 regions/autonomous provinces in Italy. (B) A detailed list of correspondences between regions/autonomous provinces and nodes.

Based on the above work, an undirected and edge-weighted network that changes dynamically over time is obtained. The next step is to extract the minimum spanning tree from the dynamic region network at each moment. In detail, when Italy is on the $t$ day of the COVID-19 pandemic, we could extract the minimum spanning tree $MST_t = (V, E)$ to better describe the evolution of Italian regional network with the change of daily cases. In this work, a classical minimum spanning tree algorithm, the Prim's algorithm, is applied to the differential weighted network $G_t$ at a specific time $t$ to obtain its minimum spanning tree $MST_t$.

### iii. *Calculating early warning indicators $I_t$*

Then, the MST-DNM indicator $I_t$ can be obtained by calculating the weight sum of the minimum spanning tree. $I_t = \sum_{i=1}^{K} W_t^i$, where $W_t^i$ represents the weight of the edge $e_i$ of the minimum spanning tree $MST_t$ at time $t$ and $K$ is the total number of edges of the minimum spanning tree $MST_t$. The specific algorithm flow is shown in Algorithm 1.

---

**Algorithm 1** The indicator $I_t$ using Prim's algorithm

---

**Input:** The nodes of the weighted undirected graph $V$; the function $w(u, v)$ which means
     the weight of the edge $(u, v)$; the function $adj(v)$ which means the nodes adjacent to $v$.

**Output:** The sum of weights of the minimum spanning tree in the input graph, $I_t$.

1:  $I_t \leftarrow 0$
2:  choose an arbitrary node in $V$ to be the root
3:  $dis(root) \leftarrow 0$
4:  **for** each node $v \in (V - \{root\})$ **do**
5:     $dis(v) \leftarrow \infty$
6:  **end for**
7:  $rest \leftarrow V$
8:  **while** $rest \neq \Phi$ **do**
9:     $cur \leftarrow$ the node with the minimum $dis$ in $rest$
10:    $I_t \leftarrow I_t + dis(cur)$
11:    $rest \leftarrow rest - cur$
12:    **for** each node $v \in adj(cur)$ **do**
13:       $dis(v) \leftarrow \min(dis(v), g(cur, v))$
14:    **end for**
15:  **end while**
16:  **return** $I_t$

---

According to DNB theory, during the critical stage, there are two cases for the minimum spanning tree $MST_t$ at time point $t$:

- In the $MST_t$, all of the nodes are DNB members;
- In the $MST_t$, DNB and non-DNB members both exist.

For the above two cases respectively, the statistical indicator $I_t$ has significant changes as presented in Table 1. Obviously, the $MST_t$ based on the indicator $I_t$ and the edges' weight $W_t$ has the ability to monitor the dynamical process of COVID-19 spread between regions and issue a warning signal timely.

    iv. *Identifying early-warning points*

In previous studies, machine learning methods, i.e., logistic regression (*Yang et al., 2020*), have been applied to identify the appearance of critical points based on years of high-dimensional data. However, for COVID-19 which originated at the beginning of 2020, the time series data obtained is of quite small scale, which is difficult for machine learning algorithm to learn the appropriate parameters and features. Therefore, the fold-change threshold, an index of volatility, is used to detect the early-warning signal. Specifically, a 2-fold change threshold is applied to identify the significant changes of the indicator $I_t$ in our study.

## The significance of indicator $I_t$

The transmission of COVID-19 is a complicated dynamic system, which contains many biomedical and social factors. Due to the massive number of influencing factors, it is

**Table 1  Critical behaviours of the MST-DNM indicator $I_t$ for different cases.**

| Case | Nodes | $SD_t$ | $\|\Delta SD_t(i,j)\|$ | $PCC_t(i,j)$ | $\|\Delta PCC_t(i,j)\|$ | $W_t^k$ | $I_t$ |
|------|-------|--------|------------------------|--------------|-------------------------|---------|-------|
| 1 | All DNM | ↗ | ↗ | ↗ | ↗ | ↗ | ↗ |
| 2 | DNM and non-DNM | D ↗ | ↗ | D ↗ | ↗ | ↗ | ↗ |
|   |  | N → |  | N ↘ | ↗ |  |  |

**Notes.**

When the system moves from time point $t-1$ to $t$, it is approaching the critical point.

"↗" represents the increase of variables; "↘" represents the decrease of variables; "→" represents that there is no significant change in the variables.

"D" represents the DNM members; "N" represents the non-DNM members.

$SD_t$ is the standard deviation at time point $t$; $PCC_t(i,j)$ is the Pearson's correlation coefficient between two nodes $v_i, v_j$.

difficult to describe the transport dynamics in high-dimensional space mathematically. The sharp or qualitative transition of regional network from the normal state to the outbreak state corresponds to the bifurcation point in dynamic system theory (*Gilmore, 1993*). According to this theory, if the system approaches the critical point, it will eventually be confined to one-dimensional or two-dimensional space, where the dynamic system can be represented in quite simple forms. This is the theoretical basis for developing a general indicator that can describe the dynamics of COVID-19 transmission.

It's clear from the above statement that the meaning of the variables in Formula (1) is as follows: (i) $\|\Delta SD_t(i,j)\|$ can describe the differential fluctuation of cases growth in two adjacent regions compared with the previous time point. (ii) $\|\Delta PCC_t(i,j)\|$ can describe the difference of the COVID-19 interaction between two adjacent regions compared with the previous time point. Apparently, attention should be paid to the edge with larger weight. Because it means that the regions associated with this edge not only worsen their own epidemic situation, but also have a great impact on the surrounding regions. Therefore, it's obvious that the indicator $I_t$, the weight sum $W_t^k$ of all edges in $MST_t$, has the ability to observe the change of a group of weighted differential networks.

## RESULTS

### Early warning of COVID-19 outbreaks in Italy

We collect the historical data of daily cases infected by COVID-19 from February 24, 2020 to November 28, 2020 in Italy. The outbreak points of COVID-19 are defined as the peak of the daily cases.

Provided as in Fig. 4, the early-warning signals are identified through the MST-DNM model for each outbreak of COVID-19. For the first wave of COVID-19 outbreak from mid-February to mid-April, the early-warning signal was issued on March 6, which is about 15 days ahead of the outbreak point. This means that our model has successfully played an early warning role.

On 9 March 2020, the Italian prime minister Mr. G Conte announced the implementation of placing the country into lockdown to restrict the movement of people, thus reducing the possibility of human to human infection (*Chintalapudi, Battineni & Amenta, 2020*; *Remuzzi & Remuzzi, 2020*). Since the last week of March, the statistics have become consciously optimistic, and the number of daily cases has been stabilizing. For

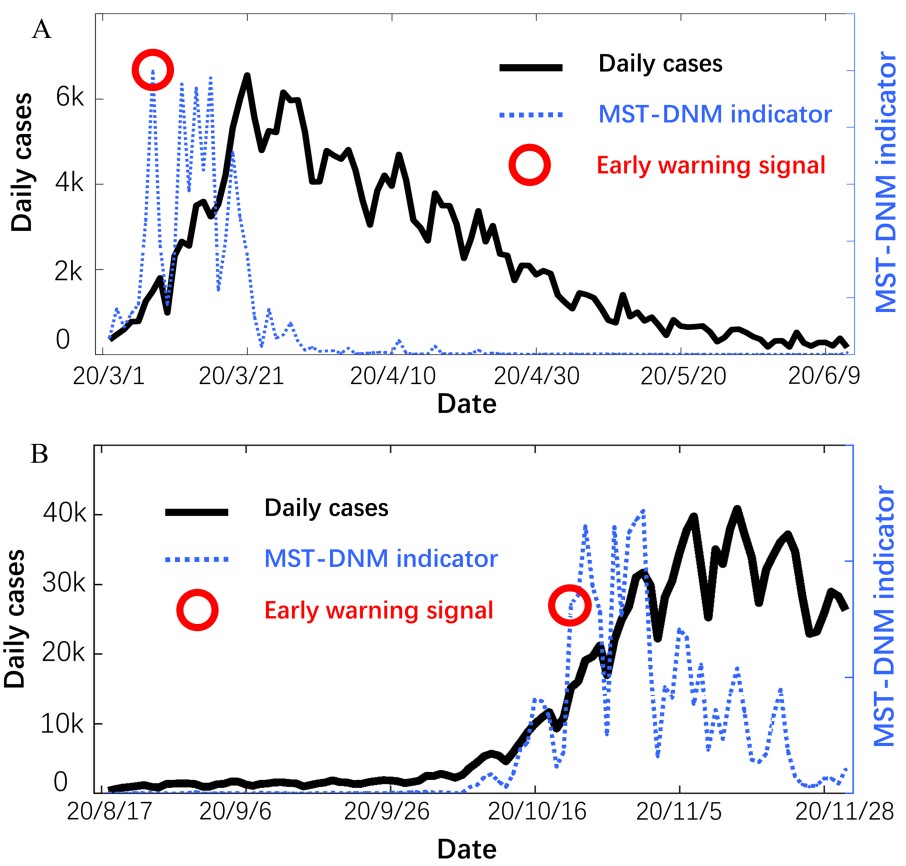

**Figure 4** **Forecast of the COVID-19 outbreaks in Italy.** In each subgraph, the left *y*-axis is the number of daily cases in Italy and the right *y*-axis is the MST-DNM indicator of corresponding date. (A) For the first wave of COVID-19 outbreak in Italy, the early-warning signal was about 15 days ahead of the outbreak point. (B) For the second wave of COVID-19 epidemic, the MST-DNM indicator significantly increases about 10 days before the actual number of confirmed cases skyrockets.

the second wave of epidemic since early October, later developing into a larger outbreak, the indicator $I_t$ was sensitive and significantly increased about 10 days before the actual number of confirmed cases skyrockets. In addition, the indicator showed a continuous downward trend with wave type after November 5, which means that the number of daily cases in Italy has initially peaked. The successful prediction of each wave of COVID-19 outbreaks in Italy demonstrates the robustness and effectiveness of the MST-DNM model in detecting real-time warning signals for emerging infectious diseases.

## The dynamics of COVID-19 transmission in Italy
### Dynamic monitoring map
To better illustrate the MST-DNM model's principle, we introduce the dynamic evolution of the COVID-19 transmission network in Italy. As shown in Fig. 5, the daily number of newly confirmed cases with MinMaxScaler in each region is mapped to each node and the correlation between two vertices of an edge is mapped to the thickness of the edge in the tree network. The specific method of MinMaxScaler is to subtract the minimum value

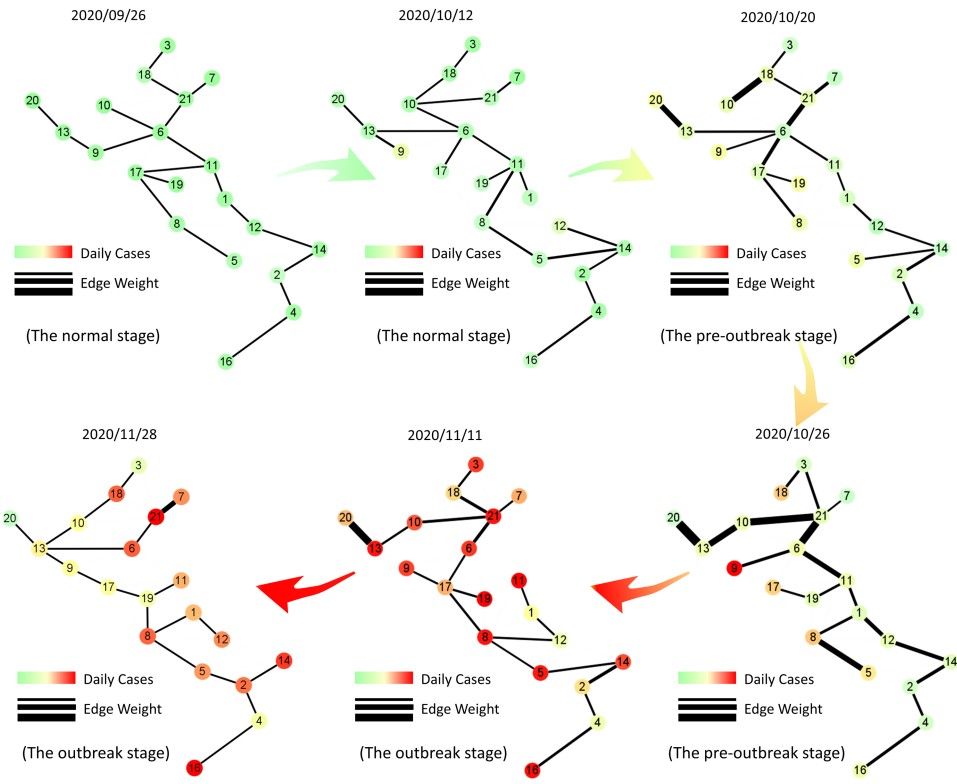

**Figure 5** **Dynamic evolution of the minimum spanning tree extracted from Italian region network.**
The tree networks are respectively selected on September 26th (the normal stage), October 12th (the normal stage), October 20th (the pre-outbreak stage), October 26th (the pre-outbreak stage), November 11th (the outbreak stage) and November 28th (the outbreak stage). In each subgraph, each node is colored by the number of daily cases with MinMaxScaler in each region and the thickness of the edge represents the correlation between two vertices of the edge. It is clear that the edges become thicker before the nodes turn darker, which indicates our model can identify the early-warning signal when the actual number of confirmed cases do not increase significantly.

of the feature from the processed value and divide it by the feature range, which is the difference between the original maximum and the original minimum. MinMaxScaler can keep the shape of the original data distribution of each region, and make the coloring of each node unaffected by other regions. It is clear that the edges became thicker before the nodes turned darker on October 20, which indicated our model identified the early-warning signal in the pre-outbreak stage when the actual number of confirmed cases did not increase significantly. After that, i.e., on October 26, the edges continued to become thicker, which meant that the epidemic might continue to worsen in Italy.

### Warning function of MST-DNM in Italy

As of November 28, the five regions with the highest cumulative number of confirmed cases in Italy are Lombardy, Piedmont, Campania, Veneto and Emilia-Romagna, which correspond to Nodes 10, 13, 5, 21 and 6 in Fig. 6A, respectively. Among them, Lombardy

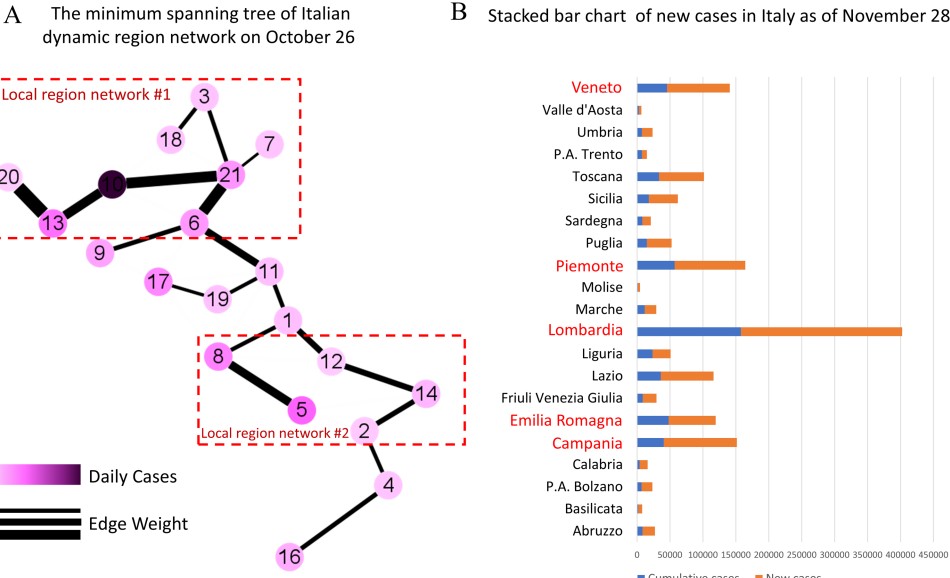

**Figure 6 Demonstration of early warning function in Italy.** (A) The minimum spanning tree of Italian dynamic region network on October 26th, colored by the scaled value of daily cases and the weight of edges, is divided into two local networks according to the thickness of the edges. (B) The stacked bar chart of new cases in Italy shows the severity of the COVID-19 outbreak after the early warning, where the blue columns represent the cumulative number of confirmed cases in each regions/ autonomous provinces in Italy as of October 26, and the orange columns represent the total number of new confirmed cases from October 27 to November 28. In particular, the red regional names highlight the five regions with the most confirmed cases. In combination, it is clear that two high-risk regions are identified by the minimum spanning tree extracted from the dynamic region network.

region is considered as the epicenter of COVID-19 outbreak in Italy (*Grasselli, Pesenti & Cecconi, 2020*; *Tuite et al., 2020*).

As shown in Fig. 6A, the dynamic region network is divided into two local networks according to the thickness of the edges, which are centered on Node 10 (Lombardy) and Node 5 (Campania) respectively. It's obvious that our model has successfully warned two high-risk regions. In fact, a large number of new cases have been confirmed in these regions over the next month as shown in Fig. 6B. In addition, the nodes sandwiched between two local networks, i.e., Node 17 (Tuscany) and Node 19 (Umbria), should also be focused on. It turns out that from October 26 to November 28, the growth rate of new case in Tuscany and Umbria exceeded 200%.

As described in Table 2, the thicker edges in two local networks on October 26, such as $e(v_{10}, v_{21})$, $e(v_{6}, v_{21})$, $e(v_{13}, v_{20})$, $e(v_{5}, v_{8})$ and $e(v_{12}, v_{14})$, should be focused on. As of November 28, the number of confirmed cases in the regions, like Lombardy, Campania, Lazio and Piedmont, corresponding to the nodes associated with these edges has a high growth rate and a large number of new cases. This also verify the early warning function of MST-DNM, which not only measures its own epidemic situation, but also reflects the regional impact on the surrounding areas.

**Table 2** Description of the early warning function of edge weight.

| Local region network # | Edge | Related regions | Newly added cases | Growth rate |
|---|---|---|---|---|
| 1 | $e(v_{10}, v_{21})$ | Lombardy | 244,726 | 154.96% |
| | | Veneto | 95,506 | 210.07% |
| | $e(v_6, v_{21})$ | Emilia-Romagna | 71,379 | 148.93% |
| | | Veneto | 95,506 | 210.07% |
| | $e(v_{13}, v_{20})$ | Piedmont | 107,150 | 187.46% |
| | | Valle d'Aosta | 3747 | 140.39% |
| 2 | $e(v_5, v_8)$ | Campania | 111,077 | 273.91% |
| | | Lazio | 80,262 | 223.35% |
| | $e(v_{12}, v_{14})$ | Molise | 3193 | 242.08% |
| | | Puglia | 37,341 | 249.39% |

**Notes.**

"Newly added cases" refers to the cumulative number of confirmed cases in the corresponding region from October 26 to November 28.

"Growth rate" refers to the newly confirmed cases in corresponding regions from October 26 to November 28 divided by the cumulative cases on October 26.

## Application of MST-DNM in northern Italy

As shown in Fig. 6B, several areas in northern Italy, such as Lombardy, Veneto, etc., are the most severely affected by COVID-19. Our model can be applied not only to the entire country of Italy, but also to an area. In order to verify the effectiveness of our model, it has also been applied to identify the early-warning signals of COVID-19 outbreaks in northern Italy. The results are presented in Figs. S2 –S3 of Supplementary Information.

## Performance comparison

The machine learning algorithms are also used to forecast the COVID-19 epidemic situation (*Parhusip, 2020*; *Singh et al., 2020*; *Parbat & Chakraborty, 2020*). Regarding the identification of early warning signals of COVID-19 outbreaks as a binary classification problem, we compare the performance of our combined model with the support vector machine (SVM). The AUC of MST-DNM is 0.9318, while that of SVM is 0.9076. It's clear that the performance of a system based on MST-DNM is better than that based on SVM when only the data of daily confirmed cases is provided. In addition, the SVM model issue an early warning signal on September 28, 2020, which is too early to be of practical significance for the second wave outbreak; and our adaptive model can actually issue an early warning signal about 10 days before the actual number of confirmed cases skyrockets. Actually, compared with traditional machine learning algorithms, the MST-DNM model has the following internal strengths. First of all, it is a model-free approach without any training and testing processes. There is no feature selection in MST-DNM strategy, which solely depends on the statistical conditions of our model. Second, it's noted that there is no limitation of the data sample size for our approach, which means that our model could achieve a good performance with only small sample data. Therefore, it can be applied to describe and monitor the emerging infectious diseases like COVID-19. In addition, our combined model is capable to describe the dynamic process of the spread of COVID-19 through the minimum spanning tree of dynamic region networks.

## DISCUSSION

A new wave of COVID-19 epidemic is sweeping the world. On November 26, more than 1.5 million people were diagnosed with COVID-19 infection in Italy, and the death toll rose rapidly in the second wave of COVID-19 epidemic, bringing a heavy burden to hospitals. In order to prevent a new wave of COVID-19 pandemic or at least reduce the magnitude of COVID-19 outbreaks, it is essential to build a surveillance system that relies entirely on reliable and available information, such as the number of daily cases.

Specifically, unlike the critical transformation analysis based on DNB of complex diseases with genomic datasets, the DNB method has been improved and applied to the macro regional networks. The successful application in Italy shows that the MST-DNM is a model-free method with data-driven characteristics and has great potential in actual real-time monitoring for emerging infectious diseases. Moreover, this is the first time that the improved method based on DNB has been applied to predict the outbreak of COVID-19. Unlike previous studies that used the DNB based methods to predict influenza outbreaks (*Yang et al., 2020*; *Chen et al., 2019*; *Chen et al., 2020*), our study is based on small time series samples rather than years of time series data. Therefore, it could be employed to describe and monitor emerging infectious diseases like COVID-19. In addition, this paper introduces the practical significance and early warning function of the MST-DNM model in detail. It is believed that this is an important step from theory to practice. It should be noted that the MST-DNM model in our work is completely based on the records of confirmed cases per day, and has achieved satisfactory performance. Given more information on the spread of the COVID-19 epidemic, the monitoring model is expected to reliably forecast the transmission and outbreak of COVID-19 in terms of sensitivity and accuracy.

Although the proposed model have achieved good results, there are some limitations of the project:

- The MST-DNM model in our work is completely based on the records of daily cases. If we could get data on the number of people tested in a region, we could measure the epidemic situation in this region more accurately. This may be the direction of model improvement in the future.
- As for the recognition of early warning signals, we can take into consideration any alternative to that criterion and the effect that a different choice could have on the prediction results in the future work.
- Experiments in this paper were performed on COVID-19 outbreaks in Italy. The future work could involve to exam the proposed model on other regions or countries.

## CONCLUSIONS

In this study, we developed a combined model with dynamic network marker and minimum spanning tree solely based on the daily cases to describe and forecast the COVID-19 outbreaks in Italy. In order to put theory into practice, we also explain the significance and warning function of the model indicators in detail. By extracting the minimum spanning tree from the dynamic region network, the model can effectively identify the early-warning

signals with an average of 2-week window lead prior to the catastrophic transition into COVID-19 outbreaks in Italy. Through the study of the network dynamics in Italy, this paper reveals the spread of COVID-19 on the network level. It is noteworthy that the driving force of MST-DNM only relies on small samples, rather than multi-year data. Therefore, it has great potential to monitor emerging infectious diseases timely.

## ACKNOWLEDGEMENTS

We would like to thank Professor Sheng Bi and Mr. Yingqi Chen for productive discussions.

### Funding

This work was supported by the National Natural Science Foundation of China (Nos. 11771152, 61703168, 11901203, 11971176, 12026608), the Guangdong Basic and Applied Basic Research Foundation (Nos. 2019B151502062, 2021A1515012317), the Guangdong Science and Technology plan project (No. 2020A0505100015) and the China Postdoctoral Science Foundation funded project (Nos. 2019M662895, 2020T130212). The funders had no role in study design, data collection and analysis, decision to publish, or preparation of the manuscript.

### Grant Disclosures

The following grant information was disclosed by the authors:
National Natural Science Foundation of China: 11771152, 61703168, 11901203, 11971176, 12026608.
Guangdong Basic and Applied Basic Research Foundation: 2019B151502062, 2021A1515012317.
China Postdoctoral Science Foundation funded project: 2019M662895, 2020T130212.
Guangdong Science and Technology plan project: 2020A0505100015.

### Competing Interests

The authors declare there are no competing interests.

### Author Contributions

- Min Dong conceived and designed the experiments, performed the experiments, prepared figures and/or tables, and approved the final draft.
- Xuhang Zhang conceived and designed the experiments, performed the experiments, analyzed the data, prepared figures and/or tables, authored or reviewed drafts of the paper, and approved the final draft.
- Kun Yang performed the experiments, analyzed the data, authored or reviewed drafts of the paper, and approved the final draft.
- Rui Liu conceived and designed the experiments, authored or reviewed drafts of the paper, and approved the final draft.
- Pei Chen conceived and designed the experiments, analyzed the data, authored or reviewed drafts of the paper, and approved the final draft.

## Data Availability

The raw data is available in the Supplemental File.

## Supplemental Information

Supplemental information for this article can be found online at http://dx.doi.org/10.7717/peerj.11603#supplemental-information.

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
