# Peer review of "Forecasting the COVID-19 transmission in Italy based on the minimum spanning tree of dynamic region network"

_PeerJ, doi:10.7717/peerj.11603_

## Round 0.1 · original submission · Major Revisions

Please consider all the issues raised by the referees in a thorough manner.

Reviewer 1 ·

Basic reporting

The article use clear technical terms, however the presentation could largely been improved.

The article includes sufficient background and the relevant prior literature is appropriately referenced, however the introduction is overwhelming, please consider the possibility of move some blocks of text in other, more appropriated sections
The structure of the article is confusing and need to improve clarity.

Figures are of sufficient resolution, relevant to the content and appropriately described but are also largely redundant.
All appropriate raw data have been made available and the submission is 'self-contained'

Experimental design

The research aims and scope are appropriated.
The investigation has been conducted with ethical and technical standards, from publicly available dataset. Methods are sufficiently described, referenced and source code of the analysis is shared to improve the reproducibility.
The research question is relevant, meaningful, of interest and timing. However, is not clear to me how the study contributes to filling knowledge gap.

It is unclear what is the innovative prediction system compared to? And what could be the real benefit of its application.
- On line 187 authors state: “the early-warning signal was issued on March 6, which is about 15 days ahead of the outbreak point. On 9 March 2020, the Italian prime minister Mr. G Conte announced the implementation of placing the country into lockdown”, based on what you said, assuming the decision of implementing the lockdown would take zero time (which is often not the case in democratic republics, such Italy, facing the the first wave of a pandemic emergency), authors method would have give 3 days of earlier warning?
Moreover, lockdown decision could presumably be based not only on new COVID-19 cases but also on socio-demographic and economical consideration.

Here are some other major flaws that may affect the experimental design, and I would appreciate the accounting or the discussion of these points:
- on line 86 “When applied to the region networks constructed based on geographical location and traffic conditions” it is unclear if the traffic conditions you mentioned were used in this analysis. And if yes where do this information came from and where are them shown? Without this information, the system appear based exclusively on proximity.
- Authors claims (in the introduction section?!) that MST-DNM “successfully identify the early-warning signals about two weeks in advance”. How do you define the successful identification? …any identification previous to the actual the peak of the daily cases? How do this takes into account measures of spreading control such as the lockdown, which is subsequent to the early-warning signal? What would have happen to the prediction system in the hypothetical scenario of a fully efficient lockdown or the opposite, no intervention at all? It really is not clear what is the “dependent variable” you are forecasting.
- How can Sardinia be simply excluded from the model? Especially when there could be the possibility of a post-summer virus hotspot? https://www.nytimes.com/2020/09/09/world/europe/coronavirus-sardinia-italy.html. For example couldn’t be considered as connected the regions with boat way connections?
- All these speculations are based on the number of new cases. This completely ignores the number of tested people. The proportion of tested people also are different between regions, since as far as I know the budget and medical resources in Italy are allocated based on regional criteria.

Validity of the findings

Both the method and the field of its application are novel and of interest to a broad audience.
The data on which the conclusions are based are robust and publicly available in a repository from the original source.
There is no controlled experimental intervention and no claims of causation.
The conclusions are appropriately connected to the original question investigated.

However, it is not clear how the goodness of the “successful identification” of the early-warning signals are accessed? And therefor, what conclusion is supported by the results.

In the proposed method strong and tacit assumption were taken for granted (traffic information, region exclusion, screened cases...).

The conclusions are strongly in favour of the proposed method but little evidence of its benefits are described in comparison with other methods.

Additional comments

Too much redundant informations are shown in the figures/tables/supplementary. Only Figure 4 and 5 in my opinion are the best way to describe your work.

Reviewer 2 ·

Basic reporting

The paper represents a noteworthy attempt to define a standard and valid mathematical method (MST-DNM model) useful to point out early signals of actual increases in infectious disease cases, in particular Covid-19. To this end, the Authors applied the MST-DNM model to the Italian daily Covid-19 epidemic open dataset and their results appear to be promising.

Experimental design

It is an explorative observational study performed to evaluate the predictive capability of a mathematical modelling.

Validity of the findings

The findings highlighted by this study are definitely significant. However, such mathematical modelling takes in no account the statistical variability in the health data deriving from various sources.

Additional comments

Limitations and remarks pointed out should be adequately addressed in the text. Some addional computations are requested.

Annotated reviews are not available for download in order to protect the identity of reviewers who chose to remain anonymous.

---

## Round 0.2 · Minor Revisions

Please closely follow the indications given by the referees when revising the manuscript and consider all points raised.

Reviewer 1 ·

Basic reporting

From previous version:
-Presentation improved: Introduction is now easier to read and Methods are clearer.
-Figure redundancy diminished.

Experimental design

From previous version:
-advantages of the proposed methodology are now clearer, comparison with current methods are now mentioned and extensively described with text beyond the numerical aspects (although no AUC confidence interval and no DeLong or similar test results provided).
-it seemed from your response that you used the presence and type of railways from a trains map as traffic condition (which I agree can be a reliable source) but I cannot spot this information in the paper
-Sardinia concerns were appropriately addressed with the addition of a sensitivity analysis and text rephrasing
-limitation were appropriately added to discussion

Validity of the findings

From previous version:
-traffic information, region exclusion and screened cases concerns were addressed or discussed

Additional comments

The paper improved from the previous version.
All major concerns were partially or fully addressed.

Some confusion remains on ROC curves, beside the SVM comparison. There is no clear description of how these were constructed; is that sensitivity and specificity the rate of correct/false early-alert given by the MST-DNM indicator? Foreach ...day? And how is the correctness defined?

Reviewer 2 ·

Basic reporting

No comment

Experimental design

No comment

Validity of the findings

No comment

Additional comments

The Authors have addressed main remarks and issues raised by this reviewer in the Discussion section and improved English.
However, they did not provide any formal test comparing the classification performances of MST-DNM and SVM. The question was: is 0.9318 statistically greater than 0.9076? To that end the Author can apply a test to compare two AUCs, namely:
Z = (AUC1 – AUC2) / SE(AUC1 – AUC2)
where SE(AUC1 – AUC2) is the standard error of the difference.
Z is a standardized normal variable which returns the probability level (P-value) of the observed difference.
SE can be also used to compute 95%CL of the same difference, as previously requested.
Finally, it is suggested removing Figure 7: it is poorly informative and the remaining Figures are more than enough to describe the study.

---

## Round 0.3 · accepted · Accept

The authors addressed the issues raised by the referees.